# Immunosuppressive Therapy in Pediatric Kidney Transplantation: Evolution, Current Practices, and Future Directions

**DOI:** 10.3390/biomedicines13123084

**Published:** 2025-12-14

**Authors:** Mohamed S. Al Riyami, Badria Al Gaithi, Naifain Al Kalbani, Suleiman Al Saidi

**Affiliations:** Pediatric Nephrology Unit, Department of Child Health, Royal Hospital, Muscat P.O. Box 1331, Oman; badria.al-ghaithi@moh.gov.om (B.A.G.); naifain.alkalbani@moh.gov.om (N.A.K.); suleiman.alsaidi@moh.gov.om (S.A.S.)

**Keywords:** pediatric kidney transplantation, immunosuppression, tacrolimus, steroid minimization, mTOR inhibitor, induction therapy, belatacept, personalized medicine

## Abstract

Pediatric kidney transplantation (KTx) offers the best outcomes for children with end-stage renal disease (ESRD), offering dramatic improvements in survival, quality of life, growth, and developmental outcomes compared to dialysis. Modern regimens centered on tacrolimus, mycophenolate mofetil, and risk-adapted induction have substantially reduced acute rejection and improved graft survival. This viewpoint summarizes the evolution of pediatric immunosuppression, current practice trends, and emerging strategies aimed at minimizing toxicity while preserving long-term graft function. Recent data show increasing use of T-cell-depleting induction, selective application of IL-2 receptor antagonists, and gradual adoption of steroid-sparing and mTOR-based protocols. Nevertheless, progress is limited by a scarcity of pediatric randomized trials, continued reliance on extrapolated adult evidence, infection risk, long-term metabolic complications, and adherence challenges during adolescence. Insights from recent trials including steroid minimization, everolimus-based regimens, and selective Belatacept use highlight opportunities for more individualized, risk-adapted therapy. Future efforts must prioritize precision approaches supported by biomarkers, multicenter collaboration, and long-term follow-up. Overall, contemporary trends support a shift toward tailored immunosuppression that balances efficacy with safety to optimize outcomes in pediatric KTx recipients.

## 1. Introduction

Kidney transplantation remains the optimal treatment for pediatric patients with end-stage renal disease (ESRD), offering superior survival, improving growth and better neurodevelopment outcomes compared with dialysis [1]. Over the past several decades, advances in donor selection, surgical techniques, and perioperative care have transformed pediatric transplantation, with current 3-year survival now exceeding 95% and acute rejection rates reduced to around 10% [2]. Central to these achievements has been the evolution of immunosuppressive therapy. Immunosuppressive regimens have progressed from early protocols based on corticosteroids and azathioprine to sophisticated combinations involving calcineurin inhibitors, antiproliferative agents, and biologics [3,4,5]. In pediatric recipients, these advances are particularly impactful due to the need to support not only graft survival but also ongoing growth and development [5]. Despite these gains, persistent challenges remain, including infection risks (e.g., CMV, EBV, BKV), post-transplant lymphoproliferative disorder (PTLD), disease recurrence, non-adherence especially during adolescence and disparities in global access to transplantation [5].

Recent OPTN/SRTR 2022–2023 data indicate that over 800 pediatric kidney transplants are performed annually in the U.S., with continued improvements in graft survival across donor types, reinforcing the need for updated immunosuppression strategies [6].

This Viewpoint article aims to synthesize evolving trends in pediatric kidney transplant immunosuppression, integrating recent registry data with current clinical perspectives. It highlights how modern strategies particularly selective induction, steroid minimization, and adherence-focused regimens are reshaping long-term management and outcomes in children.

## 2. Historical Progression

The first successful human kidney transplant, performed in 1954 between identical twins [5,7], demonstrated the feasibility of organ transplantation without the need for immunosuppression due to genetic identity. This landmark event laid the groundwork for future procedures (Figure 1).

By the 1960s, pharmacologic immunosuppression emerged with corticosteroids and azathioprine, enabling successful transplants between genetically non-identical individuals and deceased donors [7]. For children, these early advances extended survival beyond dialysis dependency for the first time, although growth impairment and chronic rejection remained significant challenges.

The 1980s, cyclosporin was introduced, a breakthrough that doubled long-term graft survival in children compared with earlier regimens and allowed for more consistent growth and school attendance [8]. However, increased immunosuppression also led to recognition of complications such as post-transplant lymphoproliferative disorder (PTLD), first documented during this period [9].

During the 1990s, tacrolimus provided superior rejection control and MMF replaced azathioprine in many centers [10,11,12]. These agents transformed pediatric care by reducing acute rejection rates to below 15%and extending graft survival beyond a decade for many children. This era also saw the first successful pediatric multi-organ transplant [13].

From 2000s onward global access to transplantation improved with new national programs established in many countries [5,14,15]. Innovations such as paired kidney exchange, the use of non-heart-beating donors and ABO incompatibility programs increased organ availability, including for pediatric recipients. This development collectively laid groundwork for toddy’s era of personalized, risk-adapted immunosuppression, enabling most children to achieve acceptable graft survival rates [2,5].

## 3. Mechanisms of Immunosuppressive Agents

Immunosuppressive therapy targets critical steps in T-cell activation to prevent graft rejection. Induction agents such as anti-thymocyte globulin, basiliximab, and alemtuzumab provide early lymphocyte depletion or IL-2 receptor blockade [16,17]. Maintenance therapy typically combines corticosteroids, calcineurin inhibitors, and antimetabolites [16]. Corticosteroids suppress pro-inflammatory cytokine transcription [18], calcineurin inhibitors block IL-2 production via inhibition of the calcineurin–nuclear factor of activated T-cells (NFAT) pathway [16,19], antimetabolites impair lymphocyte proliferation by inhibiting de novo purine synthesis [16,20], and mammalian target of rapamycin (mTOR) inhibitors prevent IL-2–driven cell-cycle progression [16,21]. Belatacept provides costimulatory blockade by inhibiting CD28-mediated signaling [16] (Table 1).

## 4. Adverse Effects of Immunosuppressive Therapy

Each class of immunosuppressive agents carries a distinct profile of side effects that must be considered in therapy selection. Tacrolimus is associated with nephrotoxicity, neurotoxicity, and a high risk of post-transplant diabetes [22]. Cyclosporine shares nephrotoxic potential but also causes hypertension and dyslipidemia [22]. MMF is frequently linked to gastrointestinal discomfort, leukopenia, and teratogenic effects [23]. mTOR inhibitors often induce hyperlipidemia, delayed wound healing, and dermatologic changes [21]. Steroids are known for their detrimental effects on growth, bone density, and metabolic homeostasis [24] (Table 2). Understanding these risks is essential for tailoring regimens to the individual needs and vulnerabilities of pediatric patients. 

In pediatric patients, these adverse effects have additional implications. Steroid-related growth suppression may affect adult height, and early metabolic complications such as dyslipidemia or diabetes may increase long-term cardiovascular risk. These age-specific considerations are essential when tailoring immunosuppressive regimens.

## 5. Trends in Immunosuppressive Regimen

Over time the landscape of immunosuppressive therapy in pediatric kidney transplantation shifted markedly. Induction immunosuppression remains nearly universal in pediatric kidney transplantation, with more than 90% of recipients receiving induction therapy through 2023 [6]. Over the past decade, the use of T-cell–depleting (TCD) agents has steadily increased, while IL-2 receptor antagonists (IL2-Ab) are now used more selectively in lower-risk patients. Combination IL2-Ab/TCD regimens account for the largest share of induction strategies, and the proportion of children receiving no induction remains low (<10%) [5]. This pattern reflects a broader shift toward tailored induction therapy based on immunologic risk rather than uniform application. Outcome studies suggest no survival advantage of rATG over IL2-RA in standard-risk recipients, but rATG carries higher risks of PTLD and viral infections, highlighting the need to balance potency with safety [25,26] (Figure 2 and Figure 3).

Regrading maintenance immunosuppressive Tacrolimus continues to dominate maintenance therapy due to its superior ability to prevent acute rejection compared with cyclosporine. According to recent OPTN data, the combination of tacrolimus + MMF + corticosteroids remain the most common regimen, used in approximately 50–60% of pediatric recipients [6]. Steroid-sparing approaches—particularly tacrolimus + MMF without steroids—have expanded gradually, while use of azathioprine, cyclosporine, and mTOR inhibitors remains low (<5%). A small proportion of cases are categorized as “other” or “none reported,” likely reflecting data-entry variability rather than true absence of therapy [6] (Figure 4).

In randomized trial comparing tacrolimus (TAC) and cyclosporin microemulsion (CyA) in renal transplantation, patient survival rates showed no statistically significant difference between the two treatment groups over a 12-month follow-up period (*p* = 0.8177) [22]. Similarly, graft survival demonstrated a non-significant trend favoring tacrolimus (*p* = 0.0704). However, the probability of remaining free from acute rejection episodes was significantly higher in the tacrolimus group compared with cyclosporin (*p* = 0.0026) [22,27]. These findings indicate that while overall patient and graft survival rates were comparable, tacrolimus provided superior protection against acute rejection during the first-year post-transplantation. Although pediatric physiology differs meaningfully from adults, evidence from adult transplantation studies continues to guide pediatric immunosuppressive practice. Because many adult trials are not repeated in children, clinicians frequently extrapolate adult data to inform pediatric decision-making. The ELITE-Symphony trial, for example, demonstrated the superiority of reduced-dose tacrolimus/MMF over cyclosporine- or sirolimus-based regimens, establishing tacrolimus/MMF as the preferred foundation in both adults and children [27]. Similarly, long-term adult data on belatacept showed improved renal function compared with cyclosporine-based therapy [28], though its uptake remains limited due to intravenous administration and cost. These findings nevertheless support the selective use of belatacept in adolescents, particularly those with adherence challenges or calcineurin-inhibitor toxicity.

Overall, the trend in pediatric renal transplantation shows increasing reliance on TCD-based induction for higher-risk patients, selective IL2-Ab use for lower-risk groups, and consistent dominance of tacrolimus-centered maintenance regimens. These shifts highlight a movement toward individualized immunosuppression aimed at balancing rejection prevention with long-term toxicity and viral infection risk.

## 6. Steroid Minimization Strategies

The pediatric population is particularly vulnerable to the side effects of corticosteroids, which can impair growth and increase cardiovascular risk [24,29]. Consequently, strategies to reduce or eliminate steroid use have gained traction. Late steroid withdrawal protocols have been shown to improve height standard deviation scores over 27 months and reduce both systolic and diastolic blood pressure [30]. These improvements were accompanied by a decreased need for antihypertensive medications. Importantly, kidney function remained stable, with no significant difference in GFR between patients who continued steroids and those who discontinued them (*p* = 0.446). Moreover, graft survival was significantly higher in the steroid withdrawal group (*p* = 0.002) [30]. However, safety concerns persist, as evidenced by the development of PTLD in a subset of patients following withdrawal. The TWIST study further validated the benefits of steroid minimization, reporting greater height gains, particularly in prepubertal children, without increased rejection risk [31], but there are limitations in this study as discussed below.

## 7. Everolimus-Based Regimens

The CRADLE trial evaluated everolimus-based regimens in pediatric kidney transplant recipients, comparing reduced-exposure tacrolimus combined with everolimus to standard triple therapy with tacrolimus, MMF, and corticosteroids. The trial found no significant difference in rates of acute rejection, graft loss, or death between the two groups. However, the everolimus group experienced a higher discontinuation rate (21.7%), often due to tolerability issues [32]. Notably, patients in the everolimus arm who also received prophylactic antiviral therapy had the highest CMV-free survival rates (*p* = 0.015), suggesting potential advantages in certain clinical scenarios [32].

## 8. Non-Adherence and Belatacept

Adherence to lifelong immunosuppression remains a major challenge in adolescent transplant recipients. Belatacept, a co-stimulation blocker administered intravenously, has shown potential as an alternative for non-adherent patients. Some adolescents switched to belatacept experienced improved or stabilized renal function, although outcomes varied. This demonstrates the need for early identification of at-risk individuals and personalized intervention strategies. Belatacept may be especially beneficial for Epstein–Barr virus (EBV)-positive adolescents who are at lower risk of PTLD and require a nephrotoxicity-sparing regimen [33].

## 9. Toward Individualized Therapy

The future of immunosuppressive therapy in pediatric kidney transplantation relies on personalization. For patients with growth concerns, early steroid withdrawal is recommended. Children with lipid abnormalities may benefit from avoiding mTOR inhibitors, while those with recurrent viral infections such as CMV or BK virus may respond better to low dose calcineurin inhibitors combined with everolimus. In cases of post-transplant diabetes, cyclosporine or belatacept-based therapies may mitigate metabolic side effects. For adolescents with poor adherence, especially those EBV-seropositive, belatacept combined with MMF offers a promising, low-toxicity alternative that supports long-term graft success [32,34].

## 10. Limitations and Perspectives

Despite advances in pediatric kidney-transplant immunosuppression, current evidence remains constrained by several methodological and clinical limitations. A major challenge is the paucity of adequately powered randomized controlled trials (RCTs) in children, which restricts the strength of recommendations and forces clinicians to extrapolate from adult data. Trials such as TWIST, CRADLE, and early belatacept pilot studies provide important insights, yet each carries limitations that hinder broad application in routine practice [31,32,33]. The TWIST trial, although demonstrating improved growth with early steroid withdrawal, used higher-than-standard prednisone doses in the control arm, making it easier to show benefit and limiting generalizability [31]. Its cohort also consisted primarily of low-risk Caucasian children, reducing applicability to immunologically diverse or high-risk populations. Similarly, the CRADLE trial evaluating everolimus with reduced tacrolimus exposure showed acceptable short-term efficacy but reported higher discontinuation rates and slower steroid cessation than planned [32]. These constraints prevent firm conclusions regarding the optimal role of mTOR-based regimens in children. Belatacept studies are even more limited; available data stem mainly from small, retrospective cohorts of non-adherent adolescents and lack long-term controlled comparison [33,35,36]. Safety concerns, particularly PTLD risk in EBV-naïve children, further limit pediatric use.

Important methodological issues persist across pediatric transplant research. Most studies use heterogeneous inclusion criteria, combine multiple induction and maintenance strategies, and lack stratification based on immunologic risk, viral risk, or metabolic profile. Many studies also use surrogate endpoints (e.g., growth, short-term rejection rates) rather than long-term graft survival, quality of life, or biomarkers of alloreactivity. Registry data provide broader insight but are limited by confounding-by-indication and center-specific practice patterns. Looking forward, future research should aim for stratified, precision-based immunosuppression, integrating biomarkers such as virus-specific T-cell monitoring, immunologic risk scores, genomic predictors of drug metabolism, and individualized viral risk assessment. Trials should also evaluate steroid-sparing regimens in more diverse populations, define the optimal use of mTOR inhibitors, and clarify where belatacept may benefit adolescents, especially those at risk of non-adherence. Given the safety concerns surrounding EBV status and overimmunosuppression, future studies must incorporate robust viral surveillance and PTLD risk mitigation. Ultimately, progress in pediatric transplant immunosuppression will depend on collaborative multicenter RCTs, harmonized outcome measures, and long-term follow-up capable of informing individualized treatment pathways. Until such data are available, clinicians must balance limited evidence with patient-specific risk factors to guide therapy.

## 11. Conclusions

The evolution of immunosuppressive therapy in pediatric kidney transplantation reflects significant scientific progress and an increasing emphasis on individualized care. While tacrolimus and MMF remain the cornerstones of modern regimens, newer agents like everolimus and belatacept, alongside strategies for steroid minimization, offer valuable tools for improving outcomes. By aligning treatment plans with patient-specific factors including age, growth potential, comorbidities, and adherence behaviors, clinicians can better navigate the delicate balance between efficacy and safety. Continued research, collaborative trials, and adaptive clinical practice are essential for further improving graft survival and quality of life in pediatric transplant recipients.

## Figures and Tables

**Figure 1 biomedicines-13-03084-f001:**
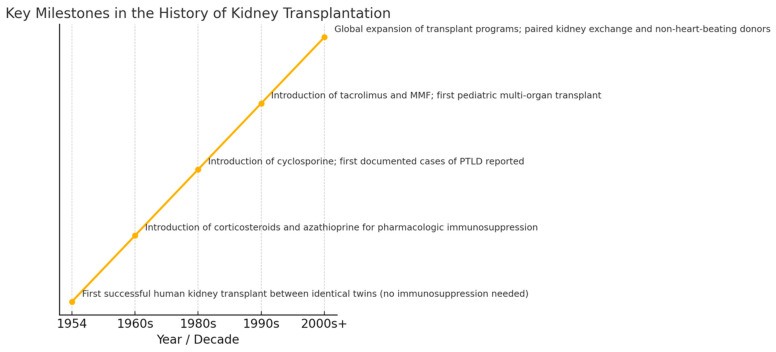
Key millstone of the history of kidney transplantation.

**Figure 2 biomedicines-13-03084-f002:**
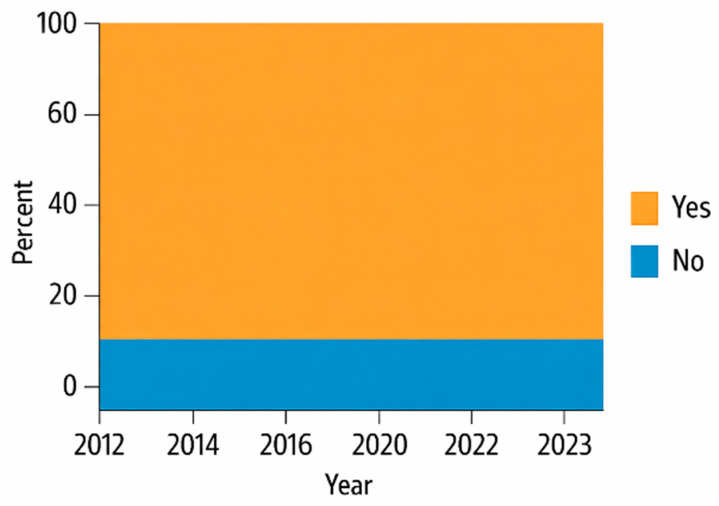
Use of induction therapy in pediatric kidney transplants OTPN 2023 report, adapted with permission from ref. [6].

**Figure 3 biomedicines-13-03084-f003:**
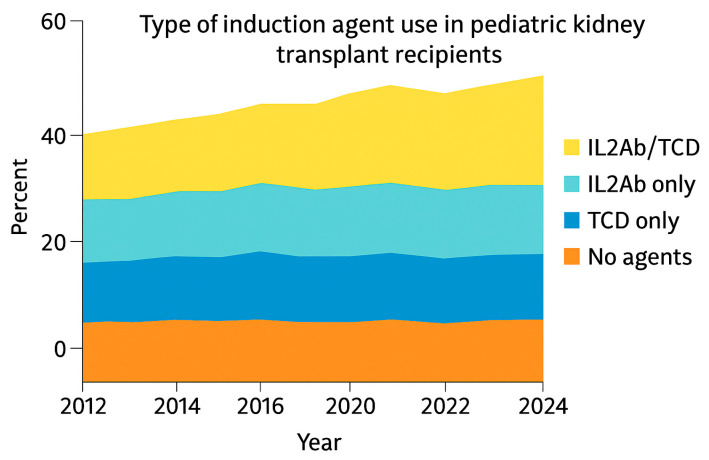
Induction agents use in pediatric kidney transplant recipients. OPTN. (2023 report); Adapted with permission from ref. [6]. Abbreviations: IL-2 = interleukin-2., TCD: T-cell depletion.

**Figure 4 biomedicines-13-03084-f004:**
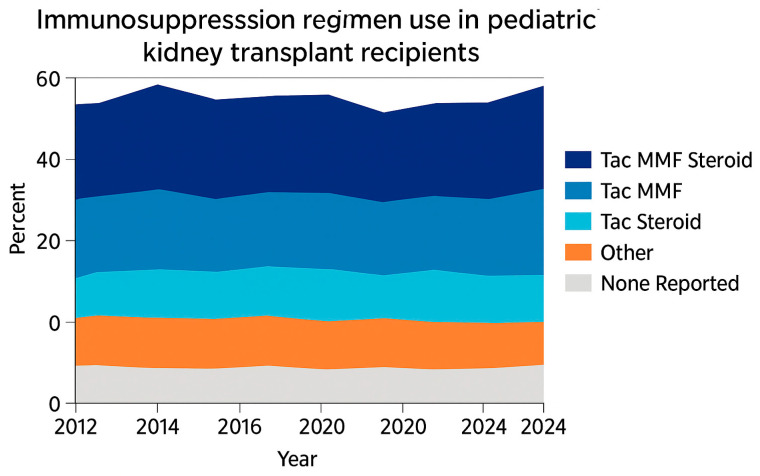
Immunosuppression regimen at transplant reported to the OPTN (2023 report) Adapted with permission from ref. [6]. Abbreviations: MMF, all mycophenolate; Tac: Tacrolimus.

**Table 1 biomedicines-13-03084-t001:** Mechanisms of Immunosuppressive Agents.

Drug Class	Examples	Mechanism of Action
Calcineurin inhibitors	Tacrolimus, Cyclosporine A	Inhibit calcineurin → block NFAT activation → suppress IL-2 transcription and T-cell activation
Antimetabolites	MMF, Azathioprine	Inhibit de novo purine synthesis → impair T- and B-cell proliferation
mTOR inhibitors	Sirolimus, Everolimus	Inhibit mTOR pathway → block IL-2–dependent cell-cycle progression
Corticosteroids	Prednisolone, Methylprednisolone	Broad cytokine suppression; genomic and non-genomic effects
Biologics	Basiliximab, Alemtuzumab; ATG	IL-2R blockade; CD52-directed depletion; polyclonal T-cell depletion
Co-stimulation blockers	Belatacept	Inhibits CD80/CD86–CD28 signaling → prevents full T-cell activation

Abbreviations: MMF = mycophenolate mofetil; NFAT = nuclear factor of activated T-cells; IL-2 = interleukin-2. ATG: Anti-Thymocyte Globulin. → = lead to.

**Table 2 biomedicines-13-03084-t002:** Adverse Effects of Immunosuppressive Therapy.

Agent	Nephrotoxicity	Hyperlipidemia	Hypertension	Diabetes	BMS
Tacrolimus	++	+	++	+++	–
Cyclosporine	+++	++	+++	+	–
MMF	–	–	–	–	+++
Sirolimus/Everolimus	+	+++	–	–	++
Glucocorticoids	–	++	+	+	–

Abbreviations: MMF, mycophenolate mofetil. BMS: Bone marrow suppression. Scale: – not typically associated; + mild; ++ moderate; +++ severe.

## Data Availability

This article does not involve data sharing, as it did not generate or analyze any new data.

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
