# Peer review of "Immunosuppressive Therapy in Pediatric Kidney Transplantation: Evolution, Current Practices, and Future Directions"

_biomedicines, 2025, doi:10.3390/biomedicines13123084_

Round 1

Reviewer 1 Report

Comments and Suggestions for Authors

The manuscript requires numerous corrections. For better readability, I will present my comments in bullet points:

Editorial corrections:

  • Correcting typos and spelling errors ("Crosponding author," "a ributed," punctuation, also in drug names: "calcineurin inhibitor," "calcinneurin inhibitor," or "Clcinneurin inhibitor")
  • Standardizing and organizing tables and figures – captions, numbering, titles.
  • Standardizing the citation style – the authors use both round brackets (e.g., "(20)") and square brackets ("[7]").

Substantive corrections:

  • The Authors cited only 34 literature sources, which is definitely too few for a review paper. Some important information was not confirmed by references at all. According to the Authors, "Each class of immunosuppressive agents carries a distinct profile of side effects that must be considered in therapy selection." Tacrolimus is associated with nephrotoxicity, neurotoxicity, and a high risk of post-transplant diabetes. Cyclosporine shares nephrotoxic potential but also causes hypertension and dyslipidemia. MMF is frequently linked to gastrointestinal discomfort, leukopenia, and teratogenic effects. mTOR inhibitors often induce hyperlipidemia, delayed wound healing, and dermatological changes. Steroids are known for their detrimental effects on growth, bone density, and metabolic homeostasis” - there are no literature references confirming this information.
  • Low scientific value: The paper does not contain new information regarding immunosuppressive therapy; it only duplicates data from other publications. In the figure titled "Trends in immunosuppression in Pediatric Kidney Transplant," the authors cite data from 2006-2018, so this data is not current.
  • It is worth adding a section “Limitations and Perspectives,” in which the authors would assess the limitations of clinical trials (e.g., CRADLE, TWIST, belatacept trials), methodological issues, and potential research directions based on the latest research.

Author Response

Reply to the reviewer comments

Thank you for considering the above article for publication in your respected journal and thank you for the comments and suggestions. Please find below the reply to reviewers’ comments.

Reviewer(s)' Comments to Author:
Reviewer: 1

1) The manuscript requires numerous corrections. For better readability, I will present my comments in bullet points:

Editorial corrections:

Correcting typos and spelling errors ("Crosponding author," "a ributed," punctuation, also in drug names: "calcineurin inhibitor," "calcinneurin inhibitor," or "Clcinneurin inhibitor")

Authors’ reply:

These were addressed and changed in the text

2) Standardizing and organizing tables and figures – captions, numbering, titles.

  • Standardizing the citation style – the authors use both round brackets (e.g., "(20)") and square brackets ("[7]").

Authors’ reply:

These were addressed and changed in the text

  • Standardizing and organizing tables and figures – captions, numbering, titles.

 Authors’ reply:

Done

  • Standardizing the citation style – the authors use both round brackets (e.g., "(20)") and square brackets ("[7]").

Authors’ reply:

Citation style was changed and standardized

Substantive corrections:

  • The Authors cited only 34 literature sources, which is definitely too few for a review paper. Some important information was not confirmed by references at all. According to the Authors, "Each class of immunosuppressive agents carries a distinct profile of side effects that must be considered in therapy selection." Tacrolimus is associated with nephrotoxicity, neurotoxicity, and a high risk of post-transplant diabetes. Cyclosporine shares nephrotoxic potential but also causes hypertension and dyslipidemia. MMF is frequently linked to gastrointestinal discomfort, leukopenia, and teratogenic effects. mTOR inhibitors often induce hyperlipidemia, delayed wound healing, and dermatological changes. Steroids are known for their detrimental effects on growth, bone density, and metabolic homeostasis” - there are no literature references confirming this information.

Authors’ reply:

     More references were added.  Also reference citations were added to the text as needed.

  • Low scientific value: The paper does not contain new information regarding immunosuppressive therapy; it only duplicates data from other publications. In the figure titled "Trends in immunosuppression in Pediatric Kidney Transplant," the authors cite data from 2006-2018, so this data is not current.

Authors’ reply:

Thank you for this valuable comment. We understand that there might be a limited new information that the article adds but we believe that readers prefer a concise and clear article were they find the information they need. Secondly, in recent years, the advancement in pediatric kidney transplantation is slower compared to earlier years. Therefore, we believe that using an older data might be understandable when there is no similar recent data. We added updated and most recent data when it is available. Due to limited pediatric studies, most of pediatric data are extrapolated from adult patients.

  • It is worth adding a section “Limitations and Perspectives,” in which the authors would assess the limitations of clinical trials (e.g., CRADLE, TWIST, belatacept trials), methodological issues, and potential research directions based on the latest research

Authors’ reply:

     This section was added in the text.

=========================================================

Reply to the reviewer comments

Thank you for considering the above article for publication in your respected journal and thank you for the comments and suggestions. Please find below the reply to reviewers’ comments.

Reviewer(s)' Comments to Author:
Reviewer: 1

1) The manuscript requires numerous corrections. For better readability, I will present my comments in bullet points:

Editorial corrections:

Correcting typos and spelling errors ("Crosponding author," "a ributed," punctuation, also in drug names: "calcineurin inhibitor," "calcinneurin inhibitor," or "Clcinneurin inhibitor")

Authors’ reply:

These were addressed and changed in the text

2) Standardizing and organizing tables and figures – captions, numbering, titles.

  • Standardizing the citation style – the authors use both round brackets (e.g., "(20)") and square brackets ("[7]").

Authors’ reply:

These were addressed and changed in the text

  • Standardizing and organizing tables and figures – captions, numbering, titles.

 Authors’ reply:

Done

  • Standardizing the citation style – the authors use both round brackets (e.g., "(20)") and square brackets ("[7]").

Authors’ reply:

Citation style was changed and standardized

Substantive corrections:

  • The Authors cited only 34 literature sources, which is definitely too few for a review paper. Some important information was not confirmed by references at all. According to the Authors, "Each class of immunosuppressive agents carries a distinct profile of side effects that must be considered in therapy selection." Tacrolimus is associated with nephrotoxicity, neurotoxicity, and a high risk of post-transplant diabetes. Cyclosporine shares nephrotoxic potential but also causes hypertension and dyslipidemia. MMF is frequently linked to gastrointestinal discomfort, leukopenia, and teratogenic effects. mTOR inhibitors often induce hyperlipidemia, delayed wound healing, and dermatological changes. Steroids are known for their detrimental effects on growth, bone density, and metabolic homeostasis” - there are no literature references confirming this information.

Authors’ reply:

     More references were added.  Also reference citations were added to the text as needed.

  • Low scientific value: The paper does not contain new information regarding immunosuppressive therapy; it only duplicates data from other publications. In the figure titled "Trends in immunosuppression in Pediatric Kidney Transplant," the authors cite data from 2006-2018, so this data is not current.

Authors’ reply:

Thank you for this valuable comment. We understand that there might be a limited new information that the article adds but we believe that readers prefer a concise and clear article were they find the information they need. Secondly, in recent years, the advancement in pediatric kidney transplantation is slower compared to earlier years. Therefore, we believe that using an older data might be understandable when there is no similar recent data. We added updated and most recent data when it is available. Due to limited pediatric studies, most of pediatric data are extrapolated from adult patients.

  • It is worth adding a section “Limitations and Perspectives,” in which the authors would assess the limitations of clinical trials (e.g., CRADLE, TWIST, belatacept trials), methodological issues, and potential research directions based on the latest research

Authors’ reply:

     This section was added in the text.

=========================================================

Reviewer 2 Report

Comments and Suggestions for Authors

The manuscript addresses an important and clinically relevant topic of immunosuppressive therapy in pediatric kidney transplantation. However, despite its educational potential, it requires substantial revision to meet the standards of a peer-reviewed journal.

  1. Title and Abstract
  • The title is appropriate, but the abstract is too generic and descriptive.
  • Please clearly state the aim and main conclusions rather than summarizing textbook knowledge.
  • Example: Lines 7–18 repeat well-known facts (e.g., “offering dramatic improvements in survival, quality of life…”) without referencing recent data or highlighting novelty.
  1. Introduction (Lines 22–37)
  • This section should better define the scope and purpose of the article. It reads like a textbook paragraph rather than the opening of a scholarly viewpoint.
  • Please specify whether this is a narrative review or opinion piece and clarify what gap in the literature you aim to address.
  • Add recent epidemiologic data (e.g., OPTN/SRTR 2022 or 2023 reports)
  1. Historical Progression (Lines 38–62)
  • The section accurately lists milestones but lacks critical commentary.
  • Suggest briefly discussing how these developments specifically affected pediatric outcomes (growth, graft survival).
  • Remove redundant references (2 and 3 both refer to early transplant history).
  • Improve figure citation: “figure 1” is mentioned but not provided.
  1. Mechanisms of Immunosuppressive Agents (Lines 64–99)
  • The text is informative but highly didactic. Consider converting the detailed drug descriptions into a concise summary table with references.
  • There are typographical errors in Table 1: Clcinneurin, Bilogics, neuclar factor of acativated T-cell.
  • Check for consistent formatting and spacing in table columns.
  • Please ensure each mechanism has a corresponding, properly cited source.
  1. Adverse Effects (Lines 100–111)
  • Table 2 is useful but poorly formatted (misaligned columns, missing % or grading scale).
  • Consider adding brief commentary linking adverse effects to pediatric considerations (growth, puberty, adherence).
  • Correct “Serolimus/Everolimus → Sirolimus/Everolimus.”
  1. Trends in Immunosuppressive Regimens (Lines 112–170)
  • This is a key section but is overly long and somewhat repetitive.
    Consolidate overlapping sentences on tacrolimus vs cyclosporine and clarify which data are derived from specific registry years.
  • Ensure all P values cited are correctly referenced (e.g., lines 127–135 from reference 23).
  • Figures 2–3 are mentioned but missing; please include them with full legends.
  • Rephrase awkward phrases: “Regrading maintainance immunosupressive medication” → “Regarding maintenance immunosuppressive medication.”
  1. Steroid Minimization and Everolimus/Belatacept Sections (Lines 171–202)
  • These sections are informative but require better linkage to pediatric evidence.
    For example, cite the TWIST and CRADLE studies with proper formatting and discuss limitations (small numbers, heterogeneous protocols).
  • Add short transitions summarizing key implications for clinical practice.
Comments on the Quality of English Language

The English language throughout the manuscript requires major editing. There are frequent grammatical, syntactic, and typographical errors (e.g., “Crosponding,” “apicable,” “Clcinneurin,” “regrading maintainance”) and inconsistent terminology for drugs and immunologic terms. Sentence structure is often awkward, with many overly long or fragmented sentences that reduce clarity.

A thorough professional English revision is recommended to improve readability, grammar, and scientific tone. The authors should ensure consistent use of medical terminology, standardized drug names, and correct punctuation and formatting according to MDPI style guidelines.

Author Response

Reviewer 2:

Reply to the reviewer comments

Thank you for considering the above article for publication in your respected journal and thank you for the comments and suggestions. Please find below the reply to reviewers’ comments.

The manuscript addresses an important and clinically relevant topic of immunosuppressive therapy in pediatric kidney transplantation. However, despite its educational potential, it requires substantial revision to meet the standards of a peer-reviewed journal.

  1. Title and Abstract
  • The title is appropriate, but the abstract is too generic and descriptive.

Authors’ reply:

     This abstract was modified as highlighted in the text.

  • Please clearly state the aim and main conclusions rather than summarizing textbook knowledge.

Authors’ reply:

     The aim was added to the text.

  • Example: Lines 7–18 repeat well-known facts (e.g., “offering dramatic improvements in survival, quality of life…”) without referencing recent data or highlighting novelty.

Authors’ reply:

This was changed and highlighted.

  1. Introduction (Lines 22–37)
  • This section should better define the scope and purpose of the article. It reads like a textbook paragraph rather than the opening of a scholarly viewpoint.

Authors’ reply:

     This section was modified as highlighted in the text.

  • Please specify whether this is a narrative review or opinion piece and clarify what gap in the literature you aim to address.

Authors’ reply:

                 This was added to the text

  • Add recent epidemiologic data (e.g., OPTN/SRTR 2022 or 2023 reports)

Authors’ reply:

  Done

  1. Historical Progression (Lines 38–62)
  • The section accurately lists milestones but lacks critical commentary.

Authors’ reply:

  Done

  • Suggest briefly discussing how these developments specifically affected pediatric outcomes (growth, graft survival).

Authors’ reply:

  This discussion was added.

  • Remove redundant references (2 and 3 both refer to early transplant history).

Authors’ reply:

  Done

  • Improve figure citation: “figure 1” is mentioned but not provided.

Authors’ reply:

  Done

  1. Mechanisms of Immunosuppressive Agents (Lines 64–99)
  • The text is informative but highly didactic. Consider converting the detailed drug descriptions into a concise summary table with references.

Authors’ reply:

  Done

  • There are typographical errors in Table 1: ClcinneurinBilogicsneuclar factor of acativated T-cell.

Authors’ reply:

  Typing errors corrected

  • Check for consistent formatting and spacing in table columns.

Authors’ reply:

  Done

  • Please ensure each mechanism has a corresponding, properly cited source.

Authors’ reply:

 Done

  1. Adverse Effects (Lines 100–111)
  • Table 2 is useful but poorly formatted (misaligned columns, missing % or grading scale).

Authors’ reply:

  The table was modified and improved.

  • Consider adding brief commentary linking adverse effects to pediatric considerations (growth, puberty, adherence).

Authors’ reply:

Added

  • Correct “Serolimus/Everolimus → Sirolimus/Everolimus.”

Authors’ reply:

Done

  1. Trends in Immunosuppressive Regimens (Lines 112–170)
  • This is a key section but is overly long and somewhat repetitive.
    Consolidate overlapping sentences on tacrolimus vs cyclosporine and clarify which data are derived from specific registry years.

Authors’ reply:

Done and highlighted

  • Ensure all P values cited are correctly referenced (e.g., lines 127–135 from reference 23).

Authors’ reply:

Done and highlighted

  • Figures 2–3 are mentioned but missing; please include them with full legends.

Authors’ reply:

Figures are added and labelled

  • Rephrase awkward phrases: “Regrading maintainance immunosupressive medication” → “Regarding maintenance immunosuppressive medication.”

Authors’ reply:

Done

  1. Steroid Minimization and Everolimus/Belatacept Sections (Lines 171–202)
  • These sections are informative but require better linkage to pediatric evidence.
    For example, cite the TWIST and CRADLE studies with proper formatting and discuss limitations (small numbers, heterogeneous protocols).

Authors’ reply:

These studies are cited and discussed.

  • Add short transitions summarizing key implications for clinical practice.

Authors’ reply:

      Added

Comments on the Quality of English Language

The English language throughout the manuscript requires major editing. There are frequent grammatical, syntactic, and typographical errors (e.g., “Crosponding,” “apicable,” “Clcinneurin,” “regrading maintainance”) and inconsistent terminology for drugs and immunologic terms. Sentence structure is often awkward, with many overly long or fragmented sentences that reduce clarity.

A thorough professional English revision is recommended to improve readability, grammar, and scientific tone. The authors should ensure consistent use of medical terminology, standardized drug names, and correct punctuation and formatting according to MDPI style guidelines.

Authors’ reply:

Quality of English language was addressed, an

Round 2

Reviewer 1 Report

Comments and Suggestions for Authors

After reviewing the revised version of the manuscript, I assess that the Authors have made many significant improvements that enhance the scientific value of the manuscript, such as the use of the latest data (OPTN/SRTR 2023–2025) and the discussion of current pediatric studies (TWIST, CRADLE, belatacept in the adolescent population). The Authors have also added a “Limitations and Perspectives” section, in which they highlighted a very important issue: the lack of robust pediatric studies and the shortage of long-term data, which results in the absence of standardized evidence-based pediatric protocols and the frequent need to extrapolate data from adult studies. At the same time, the Authors point out that the future lies in personalized therapy, improved monitoring, and further research on new immunosuppressive strategies. The conclusions presented are consistent with the evidence and arguments provided. However, extensive editorial revisions are necessary, such as organizing the content and improving the tables, figures, and references.

Author Response

After reviewing the revised version of the manuscript, I assess that the Authors have made many significant improvements that enhance the scientific value of the manuscript, such as the use of the latest data (OPTN/SRTR 2023–2025) and the discussion of current pediatric studies (TWIST, CRADLE, belatacept in the adolescent population). The Authors have also added a “Limitations and Perspectives” section, in which they highlighted a very important issue: the lack of robust pediatric studies and the shortage of long-term data, which results in the absence of standardized evidence-based pediatric protocols and the frequent need to extrapolate data from adult studies. At the same time, the Authors point out that the future lies in personalized therapy, improved monitoring, and further research on new immunosuppressive strategies. The conclusions presented are consistent with the evidence and arguments provided. However, extensive editorial revisions are necessary, such as organizing the content and improving the tables, figures, and references.

Authors’ reply:

Done

Reviewer 2 Report

Comments and Suggestions for Authors

he revised manuscript is substantially improved. The authors have addressed nearly all major concerns from the first review, resulting in a clearer, more coherent, and scientifically sound article. The English language has been markedly polished, and the structure now aligns well with the expectations of a narrative review.

A few minor points remain for further refinement:

  1. Historical section: Although improved, you may consider slightly shortening selected sentences to enhance focus on pediatric relevance.

  2. Tables: Formatting is acceptable but may benefit from uniform column alignment; this can also be handled during copyediting.

  3. Abbreviations: Please ensure that APC, NFAT, and other immunology-related abbreviations are defined consistently at first mention.

These points are minor and do not affect the scientific quality of the manuscript.
Overall, the article is now in good condition and suitable for publication after minor editorial adjustments.

Author Response

The revised manuscript is substantially improved. The authors have addressed nearly all major concerns from the first review, resulting in a clearer, more coherent, and scientifically sound article. The English language has been markedly polished, and the structure now aligns well with the expectations of a narrative review.

A few minor points remain for further refinement:

  1. Historical section: Although improved, you may consider slightly shortening selected sentences to enhance focus on pediatric relevance.

Authors’ reply:

Done and highlighted

  1. Tables: Formatting is acceptable but may benefit from uniform column alignment; this can also be handled during copyediting.

Authors’ reply:

Done

  1. Abbreviations: Please ensure that APC, NFAT, and other immunology-related abbreviations are defined consistently at first mention.

Authors’ reply:

Done